# High-Throughput Classification and Counting of Vegetable Soybean Pods Based on Deep Learning

**Chenxi Zhang, Xu Lu, Huimin Ma *** [ID]**, Yuhao Hu, Shuainan Zhang, Xiaomei Ning, Jianwei Hu and Jun Jiao**

School of Information and Computer, Anhui Agricultural University, Hefei 230036, China
* Correspondence: huiminma@ahau.edu.cn

**Abstract:** Accurate identification of soybean pods is an important prerequisite for obtaining phenotypic traits such as effective pod number and seed number per plant. However, traditional image-processing methods are sensitive to light intensity, and feature-extraction methods are complex and unstable, which are not suitable for pod multi-classification tasks. In the context of smart agriculture, many experts and scholars use deep learning algorithm methods to obtain the phenotype of soybean pods, but empty pods and aborted seeds are often ignored in pod classification, resulting in certain errors in counting results. Therefore, a new classification method based on the number of effective and abortive seeds in soybean pods is proposed in this paper, and the non-maximum suppression parameters are adjusted. Finally, the method is verified. The results show that our classification counting method can effectively reduce the errors in pod and seed counting. At the same time, this paper designs a pod dataset based on multi-device capture, in which the training dataset after data augmentation has a total of 3216 images, and the distortion image test dataset, the high-density pods image test dataset, and the low-pixel image test dataset include 90 images, respectively. Finally, four object-detection models, Faster R-CNN, YOLOv3, YOLOv4, and YOLOX, are trained on the training dataset, and the recognition performance on the three test datasets is compared to select the best model. Among them, YOLOX has the best comprehensive performance, with a mean average accuracy (mAP) of 98.24%, 91.80%, and 90.27%, respectively. Experimental results show that our algorithm can quickly and accurately achieve the high-throughput counting of pods and seeds, and improve the efficiency of indoor seed testing of soybeans.

**Keywords:** auxiliary breeding; crop phenotype; deep learning; high throughput; pod identification



## 1. Introduction

The vegetable soybean (*Glycine Max* (L.) Merrill) is a soybean harvested at approximately 80% maturity. The soybean is popular in China, South Korea, Japan, and other countries because of its high protein, fat, calcium, vitamins, and dietary fiber content [1]. In recent years, with the improvement of human health awareness and the understanding of the health function of natural products, the vegetable soybean has gradually attracted wide attention in the world. The production and trade of vegetable soybean have been rising for years [2]. In order to meet the increasing production and trade demand, it is of great significance to cultivate high-quality and high-yield vegetable soybean varieties.

As part of the soybean phenotype, the soybean pod number and seed number are powerful indicators for improving crop yield and biological research [3]. Simpson and Wilcox [4] showed that seed number and pod number are closely related to seed yield. Wang et al. [5] analyzed the variation trend of soybean varieties in 25 soybean variety areas in the Yellow-huai-hai river basin. The results showed that seeds per pod (SPPOD) and seed weight were the main reasons for the increase in yield. Liu et al. [6] identified QTLS for six yield-related traits using simple sequence repeat markers, and proposed yield-related traits including plant height, number of main stem nodes, pod number per plant, number

of seeds per pod, 100 seed weight, and number of seeds per plant. Therefore, it is necessary to develop a method to obtain pod number and seed number quickly and accurately.

At present, most of the traditional manual detection methods are time-consuming, laborious, costly, and prone to subjective errors [7]. With the development of imaging technology, image-processing technology has been widely used in crop phenotype detection [8,9]. Many scholars have achieved good results in using image-processing techniques to detect soybean phenotypes, such as screening the quantitative values of soybean seed color and various morphological characteristics [10], obtaining phenotypic information such as the length of vegetable soybeans [11], and extracting parameters related to pod color and size [12]. However, the above research requires image acquisition in a specific capture environment and sparse placement of crops without overlapping. In addition, classical image-processing technology is sensitive to texture features and the illumination conditions of objects, and has problems such as insufficient robustness and generalization ability [13,14], so it cannot carry out recognition tasks stably and effectively.

Deep learning has grown rapidly in recent years and is making a significant impact across all industries [15]. Compared with traditional digital image-processing technology, the advantage of deep learning is that the network automatically learns and extracts relevant features, which can effectively solve the problems mentioned above. Due to its powerful performance and accuracy, deep learning has become more and more popular in the field of agriculture [16], such as in fruit classification [17], crop grain recognition [18], pest and disease recognition [19], weed recognition [20], etc. In the context of smart agriculture, it is expected to use deep learning methods to classify and count soybean pods. Uzal et al. [21] proposed a classical method based on cut-clipping feature extraction (FE), the support vector machine (SVM) classification model, and Convolutional Neural Network (CNN). This method is able to estimate the pod number based on the number of soybean seeds. Riera et al. [22] used the VGG19 network model [23] to identify, isolate, and detect regions of interest in complex environments, and used tracking technology to count the number of pods in the whole plot. Yan et al. [24] used deep learning to transfer five different network models for the classification and identification of soybean pods, and the model with the highest accuracy rate reached 98.41%.

However, the traditional convolutional neural classification network can only detect a single object in a single image, which cannot meet the needs of high-throughput detection. Compared with the traditional classification network, the object-detection algorithm based on deep learning can classify and identify multiple pods in a single image, which is more suitable for high-throughput counting tasks of pods and seeds. For example, Guo Rui et al. [25] divided pods (into empty pods, one-seed pod, two-seed pod, three-seed pod, four-seed pod, and five-seed pod) and used the improved YOLOv4 [26] object-detection algorithm to achieve an average accuracy of 84.37% for these six types of pods. Xiang Yun [27] used a deep learning target-detection network and machine vision to realize the classification of vegetable soybeans and the rapid acquisition of pod length and width. Through experiment and comparison, it was found that YOLOv5 [28] was superior to the previous classification network in both single-image single-pod and single-image multi-pod classification. Zhu Ring-sheng et al. [29] compared various deep learning algorithms for identifying and counting soybean flowers and pods, and further improved and optimized the Faster R-CNN model [30] based on the characteristics of soybean flowers and pods, and the final accuracy of the model for identifying flowers and pods was improved to 94.36% and 91%, respectively. In a word, the object-detection algorithm has great potential in pod recognition.

The aim of this paper was to propose a fast and accurate pod-detection method which could solve the problem of counting pods and seeds in a soybean indoor planting test. This method can not only realize the classification and counting of pods, but also distinguish the effective grains and abortive grains inside the pod and count the effective grains separately. In this study, we first established a training dataset with 3216 images, and also made a distorted image test dataset, a high-density pods image test dataset, and a low-pixel image test dataset, containing 90 images each. Then, four object detection models, Faster R-CNN, YOLOv3 [31], YOLOv4, and YOLOX [32], were trained on the training dataset, and the performance were compared on the three test datasets to select the best model. Finally, our pod classification and counting method was validated.

The main contribution of this study are as follows:

(1) We propose a new method for fine classification of pods based on the number of effective seeds and abortive seeds in pods. This method can obtain the number of effective seeds more accurately. To the best of our knowledge, this is the first work that can distinguish the effective seeds from the abortive seeds in soybean pods.

(2) A soybean pod dataset based on multiple devices is designed to simulate the problems of high pod density, image distortion, and low resolution encountered in image acquisition in reality.

(3) The proposed object-detection network achieves good performance when processing densely sampled soybean pods, and can effectively identify distorted images and low-pixel images.

The other parts of this paper are as follows. The second part mainly includes the production of soybean pod dataset, the selection of object-detection algorithm, and the improved pod classification and counting method. In the third section, the training and testing process and analysis results are given. The last section concludes the research.

## 2. Materials and Methods

### 2.1. Materials

2.1.1. Image Acquisition and Preprocessing

The experimental materials used in this study were summer vegetable soybean pods produced in Tai'an, Shandong Province, China. The pods were randomly laid on a white background board during capture, and then an H680 scanner produced by Hanwang Technology in Beijing, China, and HonorV30pro produced by Honor Technology of Shenzhen, China, were used to capture a total of 1072 images of the pods as original images. As shown in Figure 1, the scanner was used to photograph the pods at a capture distance of 15 cm, and 710 images with a resolution of 2448 × 2448 were taken in total. Then, an HonorV30pro mobile phone installed on a fixed tripod was used to capture 362 images with a resolution of 3072 × 3072 for another batch of vegetable soybeans at a capture distance of 30 cm. The scanner can turn green pods into yellow pods, aiming to simulate the distorted images of pods caused by the capture environment, the capture equipment, or the pods themselves in real applications. Mobile phone cameras, on the other hand, are able to simulate different distances and capture more pods. The purpose of using different cameras to capture images is to improve the universality of the model, so that it can complete the detection task in different capture devices and capture environments.

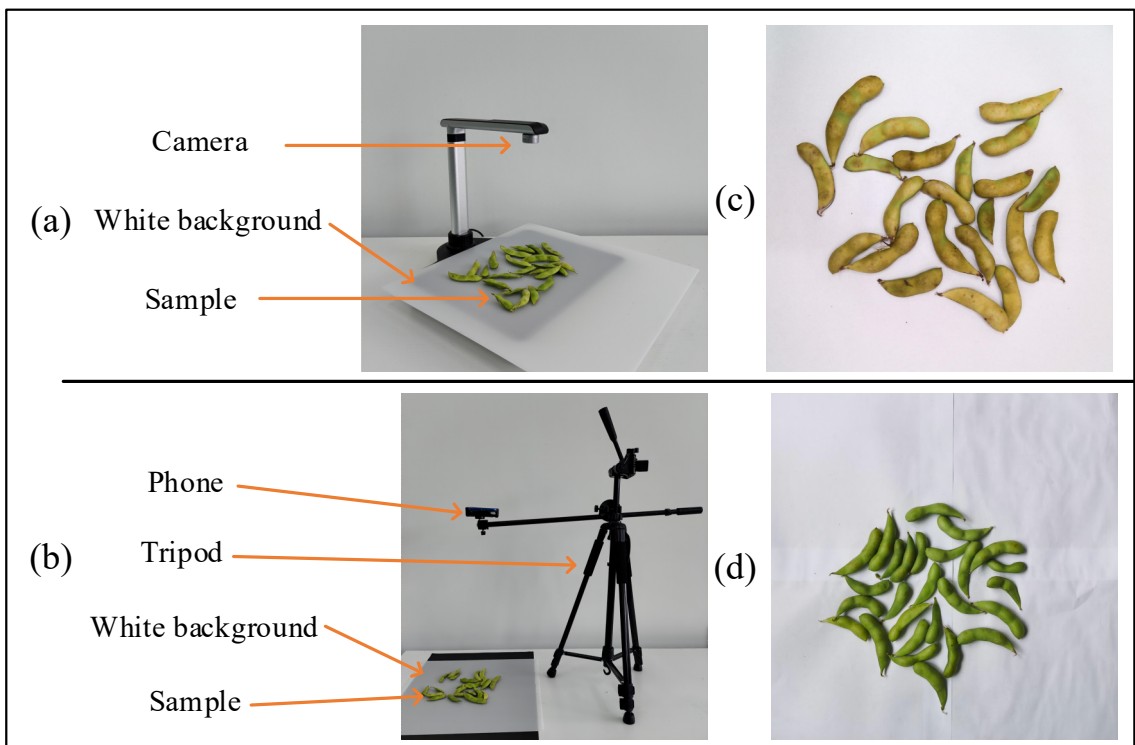

**Figure 1.** Image-acquisition platform and image-acquisition sample. (**a**) Scanner working diagram, (**b**) Mobile phone camera working diagram, (**c**) Sample image captured by scanner, (**d**) Sample image captured by mobile phone camera.

As we all know, deep convolutional neural networks need to consider many factors, one of which is the number of samples in the datasets [33]. Using data augmentation can enrich the dataset and significantly improve the performance of the model [34]. The data-augmentation method in this paper was chosen to adjust the brightness of the image to 1.25 times and 0.7 times of the original one, respectively, so as to simulate different light intensities during indoor capture. Part of the image after data augmentation is shown in Figure 2. The number of training set images is expanded to 3216 by data augmentation.

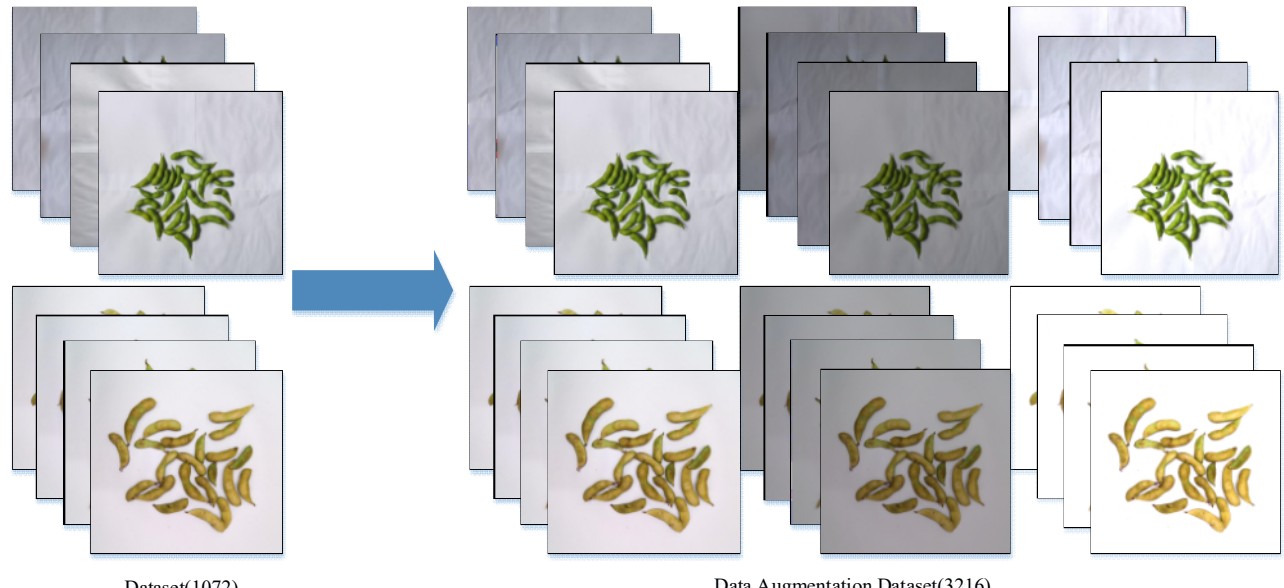

**Figure 2.** Data augmentation.

In order to ensure the independence of the test dataset, we selected another batch of vegetable soybean pods different from the training set as the capture samples and made three different test datasets to test our method. Firstly, 90 color-distorted images were taken by the scanner as the test dataset CPD (color-distorted pod image dataset), and then 90 high-density pods images were taken by Huawei P40 (produced by Huawei of in Shenzhen, China) as the test dataset HPD (high-density pod image dataset). Finally, in order to test the recognition ability of the model for low-pixel images, the 90 images of the test dataset HPD were reduced to 1024 × 1024 as was the test dataset LPD (low-pixel pod image dataset). The test dataset image is shown in Figure 3. It is worth noting that we deliberately controlled the number of pods in each image when capturing the test dataset images, so as to test the recognition performance of the model for different densities of pods. The different datasets are shown in Table 1.

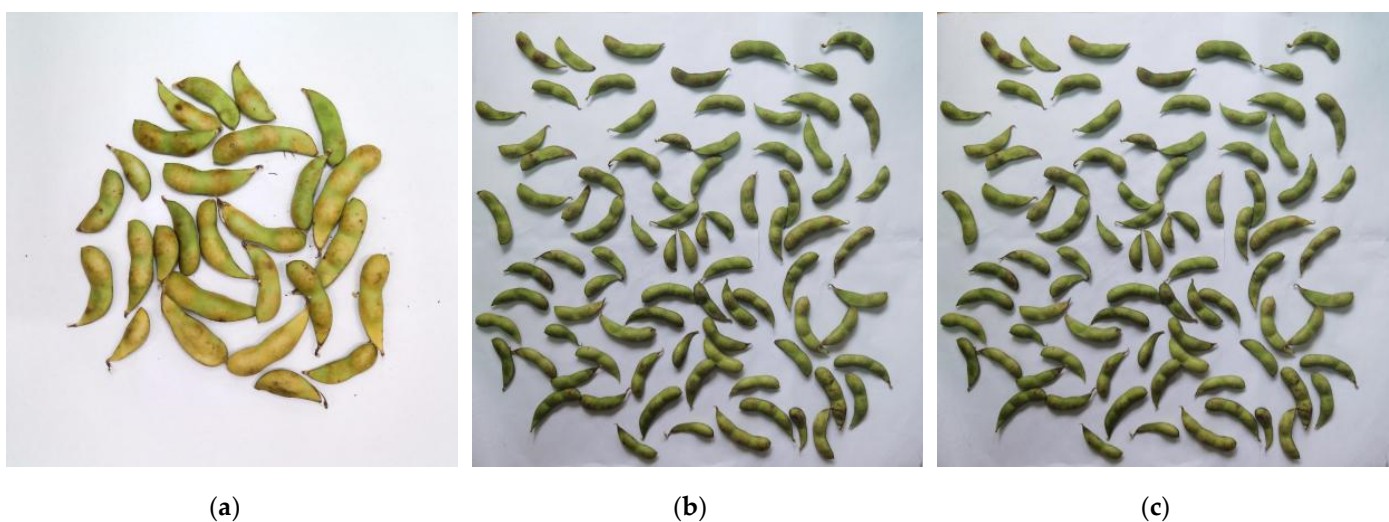

| (a) | (b) | (c) |

**Figure 3.** Samples of the soybean pods from different datasets: (**a**) test dataset CPD (color-distorted pod image dataset), (**b**) test dataset HPD (high-density pod image dataset), and (**c**) test dataset LPD (low-pixel pod image dataset).

**Table 1.** Images of different datasets.

| Dataset Type | Equipment | Image Resolution | Number of Pods | Number of Images | Total |
|---|---|---|---|---|---|
| Training Dataset | H680 Scanner [1]<br>Honor V30pro [2] | 2448 × 2448<br>3072 × 3072 | 20–50<br>30–50 | 2130<br>1086 | 3216 |
| Test dataset CPD | H680 Scanner | 2448 × 2448 | 20–50 | 90 | 90 |
| Test dataset HPD | Huawei P40 [3] | 3072 × 3072 | 20–50<br>50–80<br>80–120 | 37<br>24<br>29 | 90 |
| Test dataset LPD | Huewei P40 | 1024 × 1024 | 20–50<br>50–80<br>80–120 | 37<br>24<br>29 | 90 |

CPD represents color-distorted pod image dataset; HPD represents high-density pod image dataset; LPD represents low-pixel pod image dataset. [1] The H680 scanner was produced by Hanwang Technology in Beijing, China. [2] HonorV30pro was produced by Honor in Shenzhen, China. [3] Huawei P40 was produced by Huawei in Shenzhen, China.

### 2.1.2. Dataset Annotation

LabelImg annotation software was used to manually label the original images of the training set and the images of the test dataset, the annotation results were stored in the xml file, and then the annotation files corresponding to the data augmented images were adjusted accordingly. The different forms of pods are marked with different letters and numbers. According to the number of seeds in the pods, whether the pod is empty or not, and the number of abortive seeds, we use B to represent the number of effective seeds in the pods, and E to represent the number of abortive seeds. It is worth noting that there are three forms of empty pods; namely, a completely empty one-seed pod, a completely empty two-seed pod, and a completely empty three-seed pod, which are uniformly marked as empty in this experiment. In addition, the one-seed pod is labeled 1B, two-seed pod 2B, and three-seed pod 3B. The two-seed pod containing one abortive seed is labeled 1B-1E, three-seed pod containing one abortive seed labeled 2B-1E, and three-seed pod containing two abortive seeds is labeled 1B-2E. The classification of different pods is shown in Figure 4, and the distribution of each class of pods in the dataset is shown in Table 2.

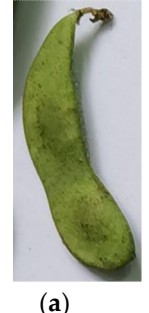 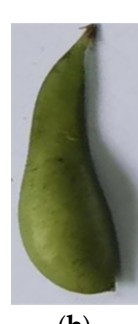 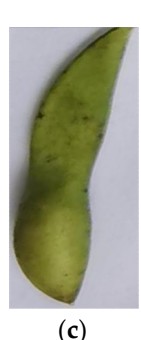 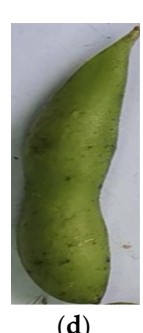 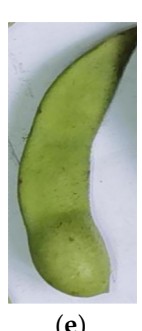 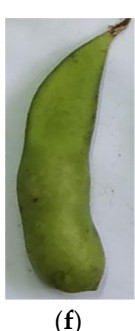 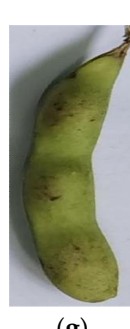

(**a**)          (**b**)          (**c**)          (**d**)          (**e**)          (**f**)          (**g**)

**Figure 4.** Samples of different pods. (**a**) Empty, (**b**) 1B, (**c**) 1B-1E, (**d**) 2B, (**e**) 1B-2E, (**f**) 2B-1E, (**g**) 3B.

**Table 2.** Distribution of each class of pods in different datasets.

| Dataset Type | Number of Pods per Class | | | | | | |
| --- | --- | --- | --- | --- | --- | --- | --- |
| | Empty | 1B | 2B | 3B | 1B-1E | 2B-1E | 1B-2E |
| Training Dataset | 1626 | 16,584 | 43,275 | 15,306 | 12,393 | 5904 | 1923 |
| Test dataset CPD | 131 | 1350 | 3723 | 1402 | 847 | 549 | 154 |
| Test dataset HPD | 101 | 811 | 2462 | 1048 | 372 | 413 | 113 |
| Test dataset LPD | 101 | 811 | 2462 | 1048 | 372 | 413 | 113 |

### 2.2. Methods

### 2.2.1. Object Detection

Object detection is divided into two categories according to detection methods: "two-stage detection" and "single-stage detection". Generally, the positioning and recognition accuracy of two-stage detection is higher, while the single-stage detection speed is faster. Among them, the two-stage object detection algorithms mainly include R-CNN [35], Fast R-CNN [36], Faster R-CNN [30], etc., while the excellent single-stage object-detection algorithms include SSD [37], YOLOv3 [31], YOLOv4 [26], and YOLOX [32].

In previous work, we have divided pods into a one-seed pod, two-seed pod, and three-seed pod, and then compared the recognition effects of SSD, YOLOv3, YOLOv4, and YOLOX, where SSD has a poor effect. Considering the recognition accuracy and recognition speed, the original network models of four object detection algorithms, Faster R-CNN, YOLOv3, YOLOv4, and YOLOX were selected without modification for comparative experiments under the same experimental environment. The four algorithms were compared in many aspects through performance evaluation indicators, and finally the algorithm with the best comprehensive performance was selected as the algorithm for pod classification and counting.

### 2.2.2. Constructing Pod Classification and Counting Method Based on Deep Learning

Figure 5 illustrates the construction process of the pod classification and counting model. Then, three test datasets are used to test the performance of the constructed model. By comparing the performance indicators of different models, the model with the best comprehensive performance is finally selected as the model of pod classification and counting. Through this model, the effective pod number, the effective seed number, and the number of each kind of pod can be obtained quickly and accurately.

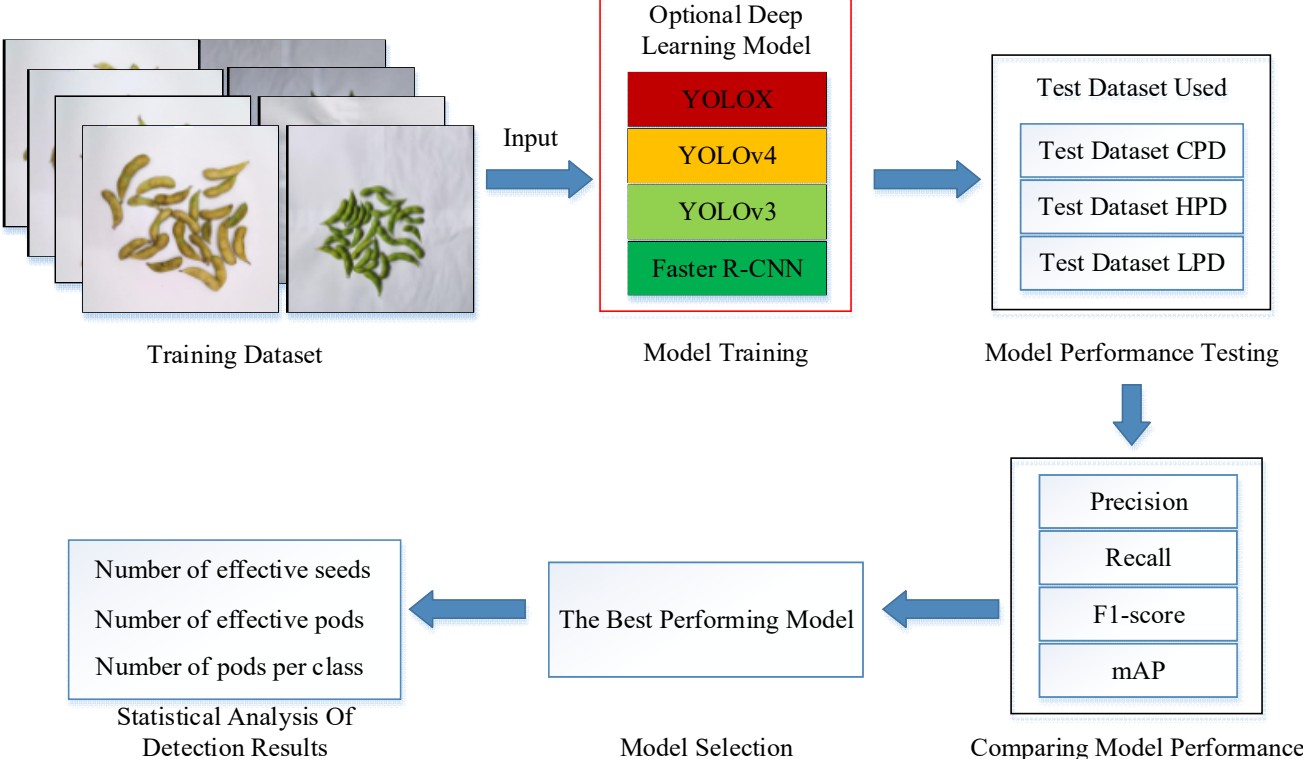

**Figure 5.** The key flow of pod classification and counting model construction.

### 2.2.3. Improved Method of Pod Classification and Counting

The number of seeds per plant refers to the number of moth-eaten, damaged, and full beans, excluding half seeds and unformed seeds, after the threshing of pods. Therefore, when estimating the number of seeds, the abortive seeds should be excluded. However, the common classification and counting method is to classify the pods according to their shapes, and classify vegetable soybeans into four categories: empty pods, one-seed pod, two-seed pod, and three-seed pod, without taking into account the presence of abortive seeds in the pods. For example, if there is an effective seed and an abortive seed in two seed pods, the abortive seed will be recognized as an effective seed if the common method is used, which will result in a large error between the number of seeds per plant and the true value. Therefore, in order to obtain the effective seed number of soybean more accurately, we use the marking method mentioned in Section 2.1.2 to classify the pods according to the number of pods, whether there are abortive seeds and the number of abortive seeds. We use B for effective seed number and E for abortive seed number, and usually a vegetable soybean pod has a maximum of three seeds, so there are seven types of pods. When estimating the seed number, we only need to count the effective seed number of each type of pod and the number of pods of each type to get an accurate count of the effective seed number. When counting effective pods, only the number of pods except empty pods needs to be counted, and the count of effective pods will not be affected. On the one hand, the subdivision of pods into 7 categories can more accurately count the number of seeds, on the other hand, it also increases the difficulty of identification.

In addition, while the counting method is usually realized by calculating the number of bounding boxes in a single detection image, considering a single soybean pod may have multiple features, a single pod may have multiple labeled boxes of different categories during the detection process. If soybean pods are detected by this method, the total number of pods often exceeds the actual number of pods, and further affects the error of seed count. In this case, a method is designed to eliminate the number of repeated computations by adjusting the non-maximum suppression parameters. The actual operation is to obtain the coordinate information of all detected bounding boxes, calculate the IoU (Intersection over Union) one by one, and set the IoU threshold to 0.2. When the IoU of two bounding boxes is greater than 0.2, it means that multiple bounding boxes appear in a single pod, and this case is removed when counting the number of pods. The IoU is calculated as shown in Equation (1).

$$\text{IoU} = \frac{\text{Area of Intersection of two boxes}}{\text{Area of Union of two boxes}} \tag{1}$$

### 2.2.4. Model Performance Evaluation Methods

In this study, classical evaluation criteria for object detection were used, including P (precision), R (recall), AP (average precision), mAP (mean average precision) and F1 score. The calculation of precision, recall, and F1 score is shown in Equations (2)–(4).

$$P = \frac{TP}{TP + FP} \tag{2}$$

$$R = \frac{TP}{TP + FN} \tag{3}$$

$$F1 = \frac{(1 + \alpha^2)P \times R}{\alpha^2(P + R)} \tag{4}$$

In the formula, P and R represent precision and recall, respectively, and TP (true positive) represents the number of positive samples detected, that is, the correct number of this type of pods can be identified. FP (false positive) represents the number of negative samples detected as positive samples, that is, the pods classified as this type of pod are actually other pods. FN (false negative) represents the number of positive samples detected as negative samples, that is, the number of pods of this type were wrongly classified into other pods. The calculation method of TP and FP quantity is as follows: firstly, the real box and the prediction box identified by the detection model are obtained, and the contents of the prediction box include the category, confidence score, and coordinate information. When the confidence score is greater than 0.5, the prediction results are retained and sorted according to the decreasing confidence score. Finally, the maximum matching IoU value between the prediction box and the real box is calculated. If it is greater than 0.5 and it is the first match, the result is denoted as TP; otherwise, it is denoted as FP. The higher the number of TP, the greater the probability of correct prediction and the stronger the detection performance of the model; otherwise, the more serious the misdetection and the lower the performance of the model. Precision is defined as the proportion of detected correct objects in all detected objects; recall rate is defined as the proportion of detected correct objects in all real objects; F1 value is used as the harmonic mean of precision and recall; here $\alpha = 1$, namely F1-score.

The average precision (AP) is defined as the area under the P-R curve, as shown in Equation (5). It is used to measure the quality of the model in each category. mAP is the result of averaging AP values of all prediction categories to measure the quality of the model in all categories. The specific definition is shown in Equation (6), where c is the total number of pod categories.

$$AP_c = \int_0^1 P_c(R_c)dR_c \tag{5}$$

$$mAP = \frac{1}{c}\sum_{i=1}^c AP_i \tag{6}$$

On the other hand, for the counting results in pods and seeds, we used $R^2$ to show the linear relationship between the test results and the true values, and used MAE (mean absolute error) and MRE (mean relative error) to measure the accuracy of the method.

The calculation of $R^2$, MAE and MRE is shown in Equations (7)–(9).

$$R^2 = 1 - \frac{\sum_{i=1}^m (y_i - \hat{y}_i)^2}{\sum_{i=1}^m (y_i - \overline{y}_i)^2} \tag{7}$$

$$R^2 = 1 - \frac{\sum_{i=1}^m (y_i - \hat{y}_i)^2}{\sum_{i=1}^m (y_i - \overline{y}_i)^2} \tag{8}$$

$$MRE = \frac{1}{m}\sum_{i=1}^m \left| \frac{y_i - \hat{y}_i}{y_i} \right| * 100\% \tag{9}$$

In the formula, m is the number of pod images to be tested, y is the true value of pods or seeds, $\hat{y}$ is the detection result of pods and seeds, and $\overline{y}$ is the average value of the true value of pods and seeds. The closer the value of $R^2$ is to 1, the better is the linear relationship between the detection result and the true value. Smaller values of MAE and MRE indicate more accurate counting results for pods and seeds.

## 3. Results

### 3.1. Model Performance Test Results

3.1.1. Comparison of Model Performance Test Results

In order to verify the recognition ability of the model, the test results of three datasets from four models (Faster R-CNN, YOLOv4, YOLOv3, and YOLOX) were compared. In order to make the model have better generalization ability and accelerate the convergence speed of the model, the experiment adopted the transfer learning method [38]. Each model iterated 150 epochs with pre-trained weights, and froze the backbone network in the first 50 epochs of training, with the learning rate set to $1 \times 10^{-4}$ and the decay rate set to 0.96. After 50 epochs, the backbone network was unfrozen, with the learning rate set to $1 \times 10^{-5}$ and the decay rate set to 0.96. After training data augmentation, 3216 pod images were segmented according to the ratio of 9:1, including 2894 training sets and 322 validation sets.

The mAP of each epoch for each of the four models was calculated separately and plotted as a mAP curve, as shown in Figure 6. Since the first 50 epochs used the transfer learning method to freeze the backbone extraction network, while the last 100 epochs unfroze the network, the curve changed significantly at the fiftieth epoch. As can be seen from Figure 6a, in the test dataset CPD, all four models can achieve good test performance, and YOLOX has the highest mAP. It can be seen from Figure 6b that in the test dataset HPD, the performance of the four models has decreased, and YOLOX has the smallest decrease, achieving better test performance. As can be seen from Figure 6c, the mAP curve in the test dataset LPD has the same upward trend as that in test dataset HPD. However, due to the decline in image resolution, the curve had an overall decline compared with that in test dataset HPD.

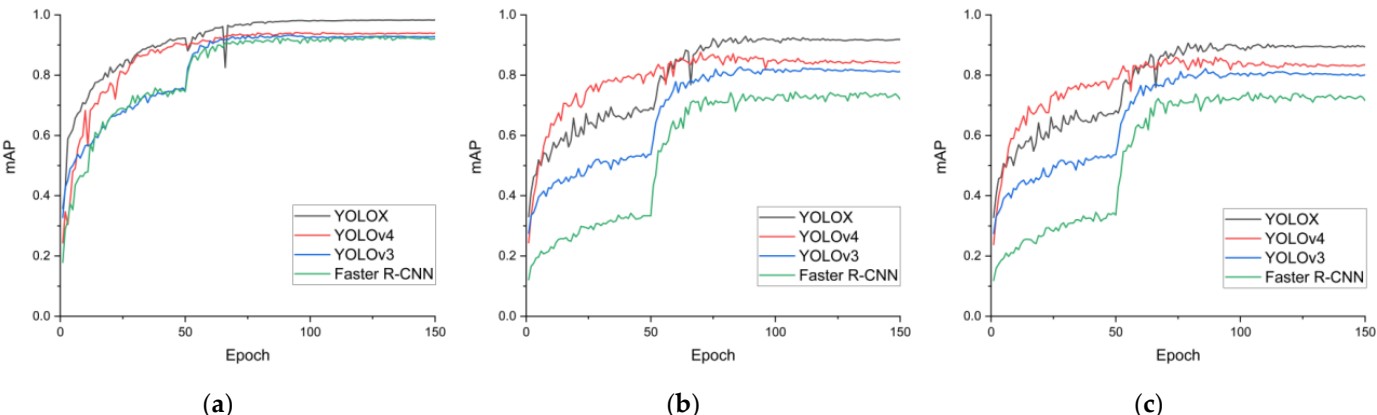

**Figure 6.** mAP curves of different models under different test datasets. (**a**) mAP curve of test dataset CPD, (**b**) mAP curve of test dataset HPD, (**c**) mAP curve of test dataset LPD.

Tables 3–5 show data obtained by different models in the test datasets CPD, HPD, and LPD, including precision, recall, F1-score, and mAP. It can be seen from Table 3 that in the test dataset CPD, the overall recognition performance of the four models was better, and the model was less affected by color distortion, which indicates that the data augmentation method (brightness adjustment) used in this paper improved the robustness of the model to some extent. Among them, YOLOX had the highest precision, recall, F1-score, and mAP. Table 4 shows the recognition effects of the four models in the test dataset HPD. It can be seen from the table that the recognition performance of different models decreased. On the one hand, because the pod images in the test dataset HPD were taken by cameras different from the training dataset, and the capture distance was longer than that in the test dataset CPD, the pod target becomes smaller, and the detail feature extraction was more difficult. On the other hand, because the maximum number of pods (120) of images in the test dataset HPD was far more than that in the test dataset CPD (50), YOLOX was least affected, and the map could still reach 92%, which indicated that the YOLOX model had stronger recognition ability in high-throughput recognition. From Table 5, we can see that in the test dataset LPD, each performance of the model decreased, which was because the image pixels in the test dataset LPD were lower and the detail features became blurred, but the mAP of YOLOX can still reach more than 90%. In summary, from the above three tables, it can be known that image color and resolution have different degrees of influence on model recognition, but the influence was less than that of the capture environment and pod density, and YOLOX was the least affected and had the best comprehensive performance in different capture environments.

**Table 3.** Comparison of test results of different models under the test dataset CPD.

| Method | P | R | F1-Score | mAP@0.5 |
|---|---|---|---|---|
| Faster R-CNN | 90.73% | 92.97% | 91.78% | 92.08% |
| YOLOv3 | 88.55% | 88.70% | 88.54% | 92.74% |
| YOLO4 | 91.18% | 90.12% | 90.62% | 93.99% |
| YOLOX | 95.54% | 97.29% | 96.39% | 98.24% |

**Table 4.** Comparison of test results of different models under the test dataset HPD.

| Method | P | R | F1-Score | mAP@0.5 |
|---|---|---|---|---|
| Faster R-CNN | 80.15% | 75.78% | 77.76% | 74.19% |
| YOLOv3 | 81.94% | 75.25% | 78.00% | 82.13% |
| YOLO4 | 88.34% | 77.29% | 90.62% | 86.30% |
| YOLOX | 85.37% | 89.36% | 86.86% | 91.80% |

**Table 5.** Comparison of test results of different models under the test dataset LPD.

| Method | P | R | F1-Score | mAP@0.5 |
|---|---|---|---|---|
| Faster R-CNN | 80.35% | 72.91% | 75.88% | 71.57% |
| YOLOv3 | 81.63% | 72.61% | 76.27% | 80.09% |
| YOLO4 | 88.11% | 73.73% | 78.62% | 83.44% |
| YOLOX | 81.67% | 87.00% | 83.77% | 90.27% |

In order to further analyze the recognition effect of the model for each class, the AP of each class of pods was calculated in the three test datasets, as detailed in Tables 6–8.

**Table 6.** Average accuracy versus average accuracy of pod category detection for different models in the test dataset CPD.

| Method | AP/% | | | | | | | MAP |
|---|---|---|---|---|---|---|---|---|
| | Empty | 1B | 2B | 3B | 1B-1E | 2B-1E | 1B-2E | |
| Faster R-CNN | 85.61% | 97.85% | 96.75% | 95.19% | 92.23% | 87.95% | 89.01% | 92.08% |
| YOLOv3 | 83.89% | 97.96% | 97.88% | 95.15% | 94.38% | 89.72% | 90.17% | 92.74% |
| YOLOv4 | 87.49% | 98.75% | 97.98% | 96.71% | 95.01% | 90.02% | 91.98% | 93.99% |
| YOLOX | 97.25% | 99.25% | 99.26% | 98.84% | 99.47% | 96.65% | 96.96% | 98.24% |

**Table 7.** Average accuracy versus average accuracy of pod category detection for different models in the test dataset HPD.

| Method | AP/% | | | | | | | MAP |
|---|---|---|---|---|---|---|---|---|
| | Empty | 1B | 2B | 3B | 1B-1E | 2B-1E | 1B-2E | |
| Faster R-CNN | 55.57% | 85.74% | 90.88% | 90.32% | 72.33% | 74.85% | 49.65% | 74.19% |
| YOLOv3 | 58.72% | 98.33% | 97.25% | 96.48% | 83.63% | 80.36% | 60.14% | 82.13% |
| YOLOv4 | 66.71% | 98.48% | 98.01% | 96.43% | 85.82% | 83.65% | 74.97% | 86.30% |
| YOLOX | 83.52% | 99.25% | 98.56% | 97.40% | 90.11% | 89.66% | 84.31% | 91.80% |

**Table 8.** Average accuracy versus average accuracy of pod category detection for different models in the test dataset LPD.

| Method | AP | | | | | | | MAP |
|---|---|---|---|---|---|---|---|---|
| | Empty | 1B | 2B | 3B | 1B-1E | 2B-1E | 1B-2E | |
| Faster R-CNN | 45.72% | 86.40% | 90.62% | 90.08% | 72.40% | 74.68% | 43.88% | 71.57% |
| YOLOv3 | 48.92% | 98.26% | 97.08% | 96.20% | 82.55% | 78.82% | 59.00% | 80.09% |
| YOLOv4 | 57.25% | 98.29% | 97.46% | 95.99% | 82.68% | 82.69% | 69.67% | 83.44% |
| YOLOX | 82.48% | 98.81% | 98.48% | 97.53% | 88.32% | 88.51% | 77.75% | 90.27% |

Table 6 shows that the four models can have good recognition performance for each class of pods under the test set CPD, and YOLOX can achieve more than 95% AP for each class of pods. It can be seen from Table 7 that in the test dataset HPD for the recognition effect of pods without abortive seeds (for the recognition effect of one-seed pods, two-seed pods and three-seed pods), except Fast-RCNN, the AP of the other three models for these three types of pods could reach more than 90%, which had a good recognition effect. Among them, the AP values of YOLOX for these three types of pods were higher than those of the other three models. For the identification of pods containing abortive seeds, YOLOX's AP of two-seed pods containing one abortive seed and three-seed pods containing one abortive seed could reach more than 90%, and the AP of the two classes of empty pods and three-seed pods containing two abortive seeds was close to 85%. However, the average recognition accuracy of YOLOv4, YOLOv3, and Faster R-CNN for pods containing abortive seeds was far worse than that of YOLOX, especially for empty pods; the average recognition accuracy of Fast-RCNN and YOLOv3 was only 55.57% and 58.72%. The average recognition accuracy of YOLOv4 was only 66.71%. As can be seen from Table 8, the performance gap between YOLOX and the other three

models will further widen when the image pixels were reduced. It can be seen from the above three tables that in different capture environments, YOLOX had stronger recognition performance for pod identification without abortive seeds, while YOLOX had an absolute advantage for pod identification with abortive seeds. This shows that the YOLOX model had higher detection performance and was more suitable for the detection of vegetable soybean pods.

### 3.1.2. Model Parameters and Performance Analysis

Because the model of Faster R-CNN series is a two-stage structure, the speed is slow, which is also reflected in the FPS values tested in Table 9. YOLO series networks use an end-to-end architecture that is faster and higher in FPS values. From the perspective of parameters, YOLOv3 had the smallest memory footprint and the highest FPS. However, considering that the research content is actually applied to pod recognition, accuracy should be the first consideration, so a small increase in memory and loss of FPS is completely acceptable for the YOLOX model when it has obtained high accuracy.

**Table 9.** Parameters and FPS of different models.

| Method | Backbone | Parameter | FPS |
|---|---|---|---|
| Faster R-CNN | ResNet50 | $2.839 \times 10^8$ | 9.08 |
| YOLOv3 | ResNet50 | $6.161 \times 10^7$ | 23.13 |
| YOLOv4 | CSPdarknet53 | $6.404 \times 10^7$ | 14.24 |
| YOLOX | CSPdarknet53 | $9.977 \times 10^7$ | 14.67 |

### *3.2. Pod Classification and Counting*

### 3.2.1. Pod Classification and Counting Test Results

In order to show the pod-recognition performance of the model more intuitively, one test image was extracted from the test datasets CPD, HPD, and LPD for recognition, and the detection results were counted so as to show the classification and counting results of pod images taken in different capture environments by the model. The identified images and detection results are shown in Figure 7.

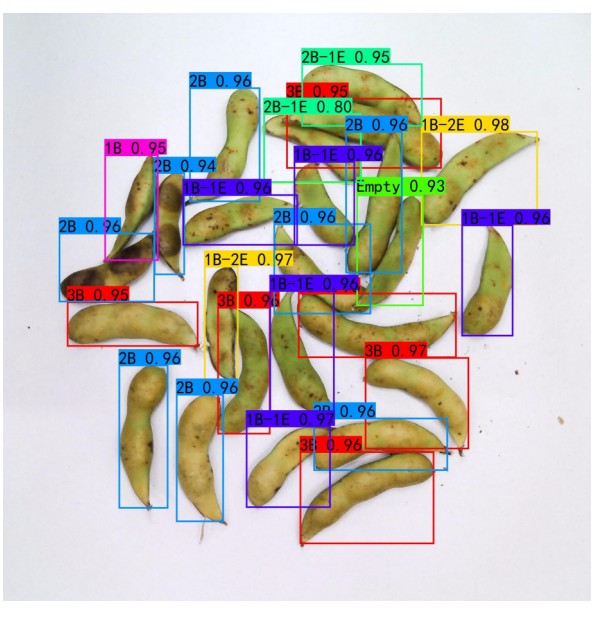

(**a**)

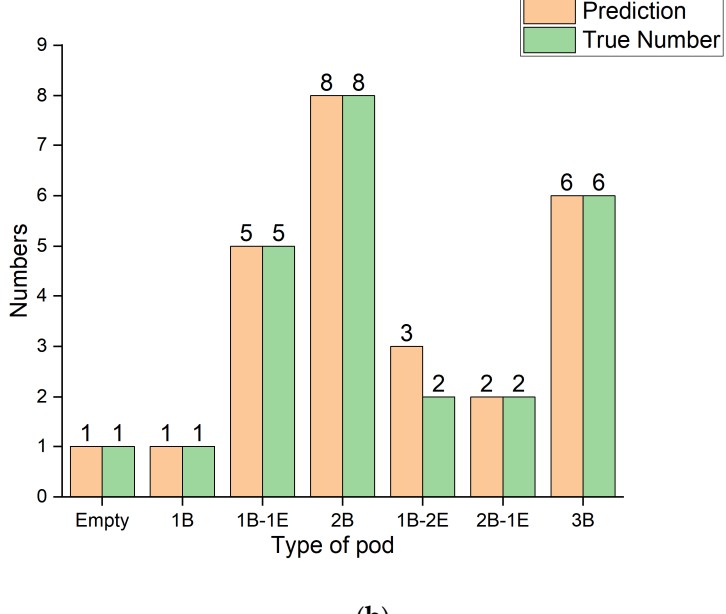

(**b**)

**Figure 7.** *Cont.*

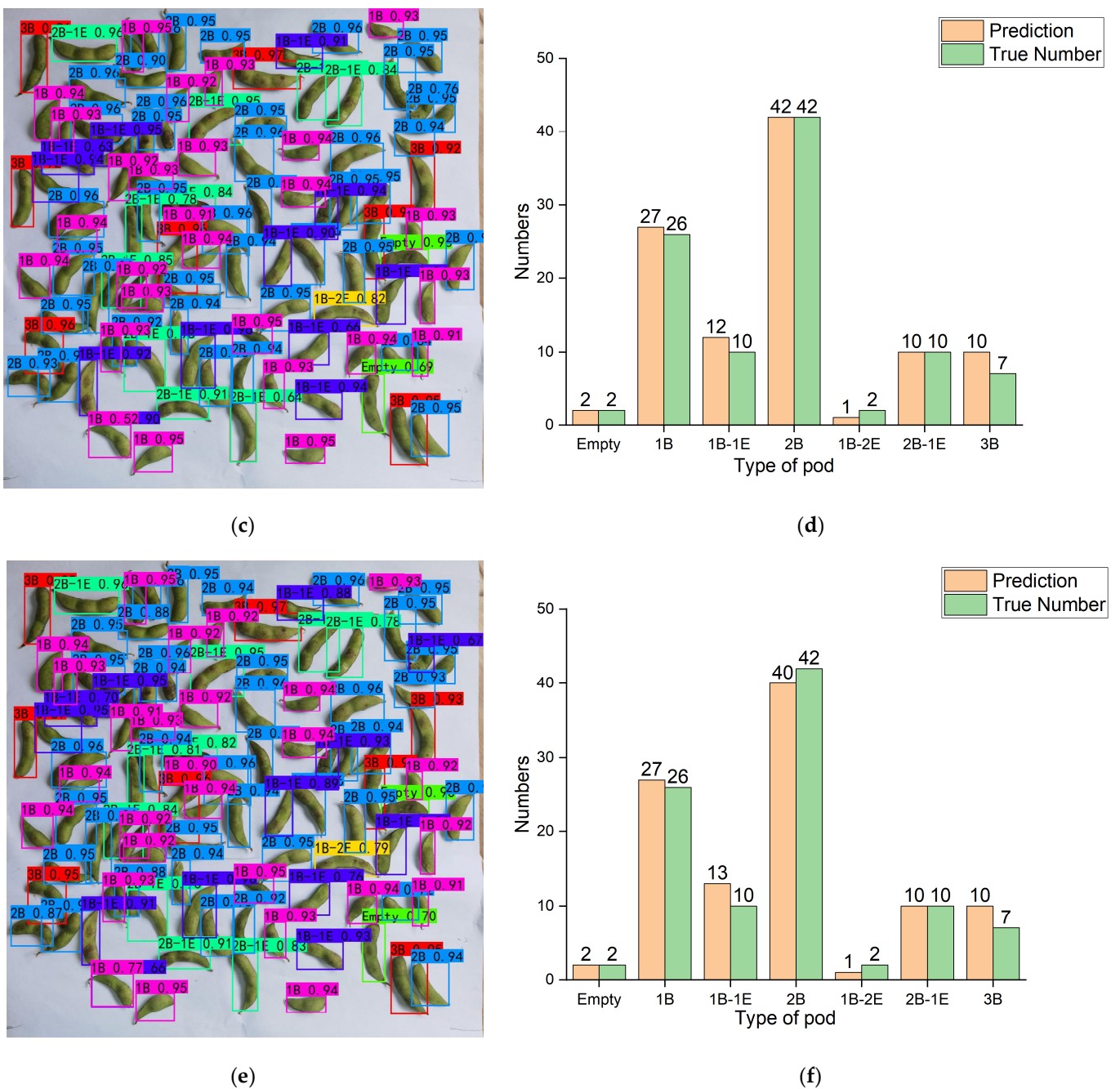

**Figure 7.** Detection and counting results of YOLOX model on example images in different test datasets. Different colored boxes are used to represent the detection results of different classes of pods. (**a**) Detection results of YOLOX on the test dataset CPD, (**b**) Count results of YOLOX on the test dataset CPD, (**c**) Detection results of YOLOX on the test dataset HPD, (**d**) Count results of YOLOX on the test dataset HPD, (**e**) Detection results of YOLOX on the test dataset LPD, (**f**) Count results of YOLOX on the test dataset LPD.

As can be seen from Figure 7b, the model can effectively identify and classify pods, which indicates that the model can still effectively solve these problems even with pod image distortion caused by the capture equipment or color change caused by the deterioration of the pod itself. In Figure 7d, in order to test the high-throughput recognition ability of the model, a high-density pod image was specially selected, and the number of pods was 99. It can be seen from the figure that when the number of pods increased to about 100, the model can still effectively identify and classify pods, while the number of

bean pods per plant was usually around 40, which fully demonstrated that the model can achieve high-throughput recognition. Finally, it can be seen from Figure 7f that even if the image resolution was reduced, YOLOX was still able to perform effective recognition, which greatly increased the applicability of the model.

On the other hand, pod adhesion and overlap were difficulties in pod recognition. Images of adhesion and overlap were selected in the dataset. Figure 7a,c,e shows that YOLOX can solve the problem of adhesion well, and even in the case of partial overlap, the model can still effectively identify each type of pod.

In summary, the model has strong recognition performance, and can solve the influence of image color and resolution size on the model, and can achieve high-throughput recognition. Only the detected pod image needs to be passed into the model to complete the classification and counting of pods, which greatly improves the detection efficiency.

### 3.2.2. Improved Classification and Counting Methods

To demonstrate the feasibility of our method, 90 pod images from the test dataset HPD were tested in this study, and the common counting method and the improved method were compared. As can be seen from Figure 8a, the results obtained by using the common counting method had a good linear relationship with the true value ($R^2$ = 99.8%), the mean absolute error was 1.98, and the mean relative error was 4.93%. As can be seen from Figure 8b, the results obtained by using the improved counting method also had a good linear relationship with the true value ($R^2$ = 99.6%), the mean absolute error was 1.61, and the mean relative error was 3.99%. The improved pod-counting method further reduces the counting error and obtains more accurate results.

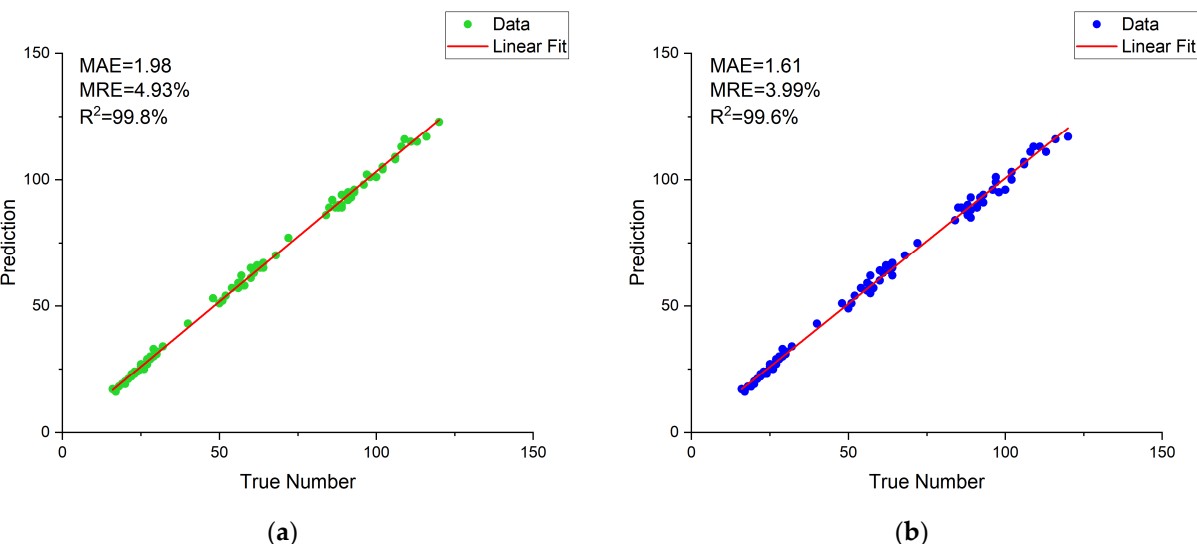

(**a**)　　　　　　　　　　　　　　　　(**b**)

**Figure 8.** Analysis of pod counting results. (**a**) Correlation analysis between count results of common counting method and true value, (**b**) Correlation analysis between count results of improved counting method and true value.

Taking Figure 7 as an example, the three images of Figure 7 were detected, the number of each class of vegetable soybean pods was counted, and then the effective number of seeds was calculated. First, we simply divided the pods into empty pod, one-seed pod, two-seed pod, and three-seed pod according to the common classification method, and calculated the effective seed number. Then the pods were classified into seven categories according to our classification method, and the number of effective seeds was counted again. Finally, the counting results of the two methods were compared with the real values, and the comparison results were shown in.

**Table 10.** Counting results of effective seeds number for vegetable soybean by different methods.

| Method | Number of Effective Seeds |
|---|---|
| Common classification method | 454 |
| Improved classification method | 392 |
| True value | 372 |

Table 10 shows that the experimental method can obtain more accurate effective seeds number compared with the common method, and when the data size increased, the advantages of the experimental method compared with the common counting method will be further expanded. In order to further prove the feasibility of the method, 90 pod images in the test dataset HPD were tested and counted according to the classification method in Section 2.2, and compared with the common classification method. The test results are shown in Figure 9.

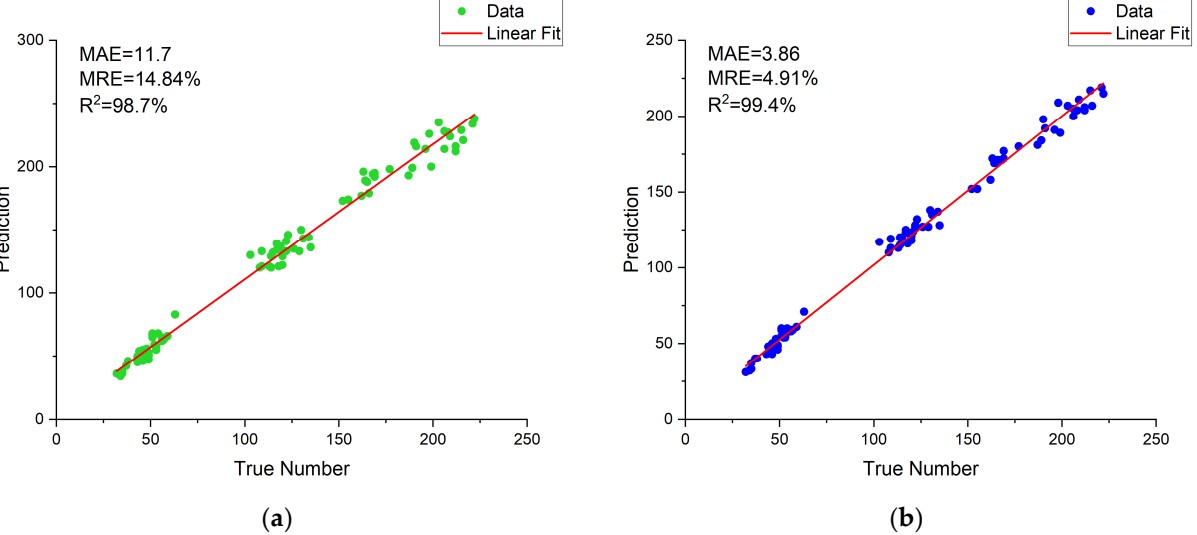

| (a) | (b) |

**Figure 9.** Analysis of seed counting results. (**a**) Correlation analysis between count results of common classification method and true value, (**b**) Correlation analysis between count results of improved classification method and true value.

The results showed that the effective number of soybean kernels measured by the common classification method had a good linear relationship with the true value ($R^2$ = 98.7%), the MAE was 11.7, and the MRE was 14.84%. The effective seed number measured by the improved counting method showed a better linear relationship with the true value ($R^2$ = 99.4%), the MAE was 3.86, and the MRE was 4.91%, which was much smaller than the common classification method, indicating that our method has higher accuracy than the common classification method, and is more suitable for counting soybean seed number.

### 3.3. Software Interaction Platform

After building the object-detection model, the software interactive platform is made based on PYQT5, so as to achieve the purpose of obtaining the detection results more intuitively and conveniently. The platform interface is shown in Figure 10. The platform mainly realizes three functions: single-image detection, video detection, and batch detection. In the main interface, there are two modes of "Image/video detection" and "Batch detection" to choose from. The "image/video detection" mode is used to detect a single image or open the camera for dynamic detection, and the phenotypic parameters such as pod number and seed number of various soybean pods in the image can be obtained in time at the left detection result. The "Batch detection" mode is used to process multiple images to be detected in batches. Enter the folder path, click the Detect button to obtain the detection

results of each image, and receive the total detection results on the right side. At the same time, you can export the batch-detection results.

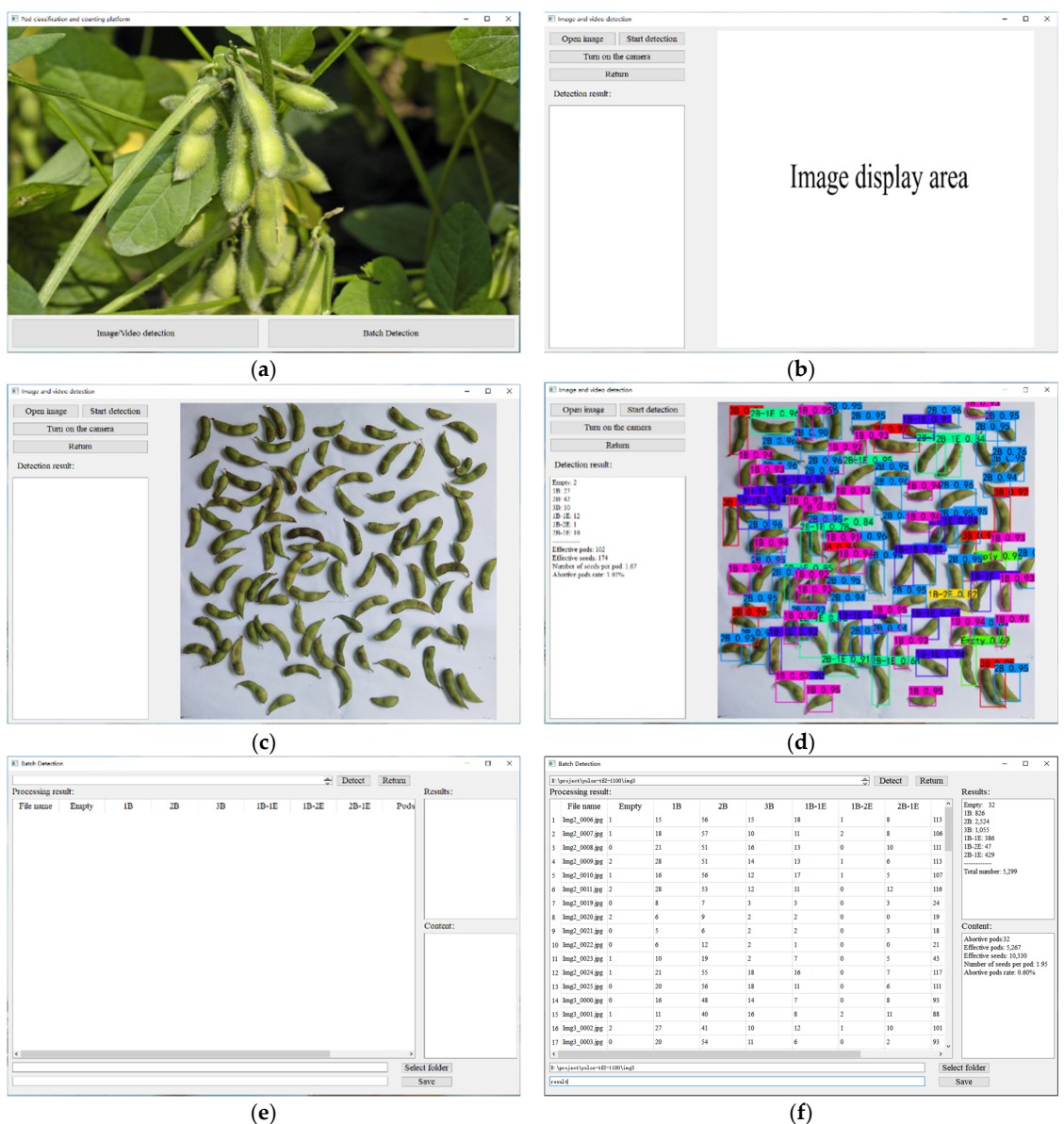

**Figure 10.** Software interaction platform. (**a**) Main interface of platform, (**b**) Single image/video detection, (**c**) Input images to be detected, (**d**) Single image detection results, (**e**) Batch detection interface, (**f**) Batch detection results and result saving.

## 4. Conclusions

The purpose of this work was to study the method of high-throughput classification and identification of soybean pods, and finally realize the accurate count of soybean seed number and soybean pod number. In order to reduce the error of seed counting caused by empty pods and abortive seeds in pods, a classification method based on the number of available seeds and aborted seeds in soybean pods was first proposed. The non-maximum suppression parameters were adjusted to reduce the pod count error. At the same time, we built a training dataset containing 3216 images, and built a distorted image test dataset, a high-density pod image test dataset, and a low-pixel image test dataset (each containing 90 images). Then, four target detection models (Faster R-CNN, YOLOv3, YOLOv4, and YOLOX) were trained on the training dataset, and the performance on the three test datasets was compared. Among them, YOLOX has the best overall performance, with mAP of 98.24%, 91.80%, and 90.27% in

test dataset CPD, test dataset HPD, and test dataset LPD, respectively. Finally, the number of pods and seeds obtained by the proposed classification and counting method was analyzed with the results obtained by common classification methods. The MRE of pod and seed count of the method was 3.99% and 4.91%, which is 0.94% and 9.93% lower than that of the common method. Experimental data show that our high-throughput pod classification and counting method speed up the laboratory testing of soybeans, solves the problem of pod and seed counting in the laboratory testing of soybeans, and reduces the errors of pod and seed counting, which is helpful to improve current and future breeding programs.

Although our method can complete pod classification and counting well, there are still some limitations at present. Firstly, pods need to be manually picked from the plant, resulting in excessive labor time. Secondly, this study focuses on indoor seed testing, which is specifically reflected in the need to randomly place pods on a white background board. In the future research, we intend to further explore a more accurate and convenient method to directly detect the complete soybean plant outdoors to obtain the category and number of pods.

**Author Contributions:** Conceptualization, C.Z. and X.L.; data curation, C.Z., X.L., Y.H., and X.N.; formal analysis, C.Z., X.L., and J.H.; funding acquisition, H.M. and J.J.; investigation, C.Z. and H.M.; Methodology, C.Z. and X.L.; project administration, H.M. and J.J.; resources, H.M. and J.J.; software, C.Z.; supervision, H.M. and J.J.; validation, C.Z., X.L., Y.H., S.Z., and J.H.; visualization, C.Z., X.L., S.Z., and X.N.; writing—original draft, C.Z. and H.M.; writing—review and editing, C.Z. and H.M. All authors have read and agreed to the published version of the manuscript.

**Funding:** This research was funded by Anhui Provincial University Excellent Young Talents Support Project, grant number gxyq2022004, Major Science and Technology Project of Anhui Province, grant number 201903a06020009, Graduate Student Innovation and Entrepreneurship Project of Anhui Province, grant number 2022cxcysj065.

**Data Availability Statement:** The data are not publicly available due to privacy restrictions.

**Conflicts of Interest:** The authors declare no conflict of interest.

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
