# Peer review of "High-Throughput Classification and Counting of Vegetable Soybean Pods Based on Deep Learning"

_agronomy, doi:10.3390/agronomy13041154_

Round 1

Reviewer 1 Report

This manuscript, "High-Throughput Classification and Counting of Vegetable Soybean Pods Based on Deep Learning" (agronomy-2231598), presents a new method for high-throughput classification and counting of soybean pods based on deep learning. Four object detection models, Faster R-CNN, YOLOv3, YOLOv4, and YOLOX, were tested on three datasets, and YOLOX showed the best results with a mean average accuracy of 98.24%, 91.80%, and 90.27%, respectively, in the three datasets. The results suggest that the new method can achieve faster and more accurate classification and counting of soybean pods, which is significant for soybean breeding. Its manuscript are interesting, but need revision and change in English grammar and style.

Abstract need reformulation, because long, unattractive and wordy! Please consider shortening the abstract by half. The abstract must be attractive and call the reader to the main text.

Introduction and Discussion is long and unattractive. Shortten were necessary. In addition, perhaps this is because a mechanistic hypothesis was not tested. When rewriting the manuscript and providing a mechanistic hypothesis in introduction and not wandering. The discussion will become more attractive as the discussion serves to validate (1) the methods, (2) the results and (3) the conclusion. Limitations of the study and sensitive points that led to the conclusion should also be validated/discussed. So please, revise the whole discussion.

-keywords in alphabetical order;

- Standardizing equipment/reagents/software nomenclature with fabricant, city, state, country (three-letter);

-Checking all manuscript for proper standardization.

Scientific names in italics;

Some figures need improve quality (Figure 9)

L118-131. Its major rewrite;

L142. 30 cm;

y-axis and x-axis were need define.

-Check old references for accuracy.

Best regards

Author Response

Responses to the comments of Reviewer #1

1.Abstract need reformulation, because long, unattractive and wordy! Please consider shortening the abstract by half. The abstract must be attractive and call the reader to the main text.

Response: Thank you very much for your suggestion, we have rewritten the entire abstract and shortened the entire abstract. At the same time, our work and the results of the paper are highlighted in the abstract. The revised content is as follows. For the convenience of your review, the text of the revised manuscript has also been marked in red font:

Accurate identification of soybean pods is an important prerequisite for obtaining phenotypic traits such as effective pod number and seed number per plant. However, traditional image processing methods are sensitive to light intensity, and feature extraction methods are complex and unstable, which are not suitable for pod multi-classification tasks. In the context of smart agriculture, many experts and scholars use deep learning algorithm methods to obtain the phenotype of soybean pods, but empty pods and aborted seeds are often ignored in pod classification, resulting in certain errors in counting results. Therefore, a new classification method based on the number of effective and abortive seeds in soybean pods is proposed in this paper, and the non-maximum inhibition parameters are adjusted. Finally, the method is verified. The results show that our classification counting method can effectively reduce the errors in pod and seed counting. At the same time, this paper designs a pod dataset based on multi-device capture, in which the training dataset after data augmentation has a total of 3216 images, and the distortion image test dataset, the high-density pods image test dataset and the low-pixel image test dataset include 90 images respectively. Finally, four object detection models such as Faster R-CNN, YOLOv3, YOLOv4 and YOLOX are trained on the training dataset, and the recognition performance on the three test datasets is compared to select the best model. Among them, YOLOX has the best comprehensive performance, with the mean average accuracy (mAP) of 98.24%, 91.80% and 90.27%, respectively. Experimental results show that our algorithm can quickly and accurately achieve high-throughput counting of pods and seeds, and improve the efficiency of indoor seed testing of soybean.

2.Introduction and Discussion is long and unattractive. Shortten were necessary. In addition, perhaps this is because a mechanistic hypothesis was not tested. When rewriting the manuscript and providing a mechanistic hypothesis in introduction and not wandering. The discussion will become more attractive as the discussion serves to validate (1) the methods, (2) the results and (3) the conclusion. Limitations of the study and sensitive points that led to the conclusion should also be validated/discussed. So please, revise the whole discussion.

Response: Thank you very much for your advice. We have revised and streamlined the introduction. Meanwhile, we rewrote lines 118-131 of the original manuscript to introduce the purpose of this paper and the working process. As stated in the answer to question 8. Finally, we add the contribution of this paper and the introduction to the structure of the paper in line 6-20 of page 3 of the revised manuscript. The specific content added is as follows, which is also marked in red in the text of the revised manuscript:

The aim of this paper is to propose a fast and accurate method for pod detection. This method can not only realize the classification and counting of pods, but also distinguish the effective grains and abortive grains inside the pod and count the effective grains sepa-rately. In this study, we first established a training dataset with 3216 images, and also made a distorted image test dataset, a high-density pods image test dataset, and a low-pixel image test dataset containing 90 images each. Then, four object detection models such as Faster R-CNN, YOLOv3, YOLOv4 and YOLOX were trained on the training da-taset, and the performance were compared on the three test datasets to select the best model. Finally, our pod classification and counting method was validated.
The main contribution of this study are as follows:
(1) We propose a new method for fine classification of pods based on the number of effective seeds and abortive seeds in pods. This method can obtain the number of effective seeds more accurately. To the best of our knowledge, this is the first work that can distin-guish the effective seeds from the abortive seeds in soybean pods.
(2) A soybean pod dataset based on multiple devices is designed to simulate the problems of high pod density, image distortion and low resolution encountered in image acquisition in reality.
(3) The proposed object detection network achieves good performance when pro-cessing densely sampled soybean pods, and can effectively identify distorted images and low-pixel images.
The other parts of this paper are as follows. The second part mainly includes the production of soybean pod dataset, the selection of object detection algorithm, and the im-proved pod classification and counting method. In the third section, the training and test-ing process and analysis results are given. The last section concludes the research.

3.keywords in alphabetical order;
Response: Thank you very much for your advice. We have reordered the keywords by first letter. Specific modifications are as follows:
Auxiliary breeding; Crop phenotype; Deep learning; High throughput; Pod identification;

4.Standardizing equipment/reagents/software nomenclature with fabricant, city, state, country (three-letter);

Response: We have standardized the naming of equipment and raw materials in the article, and the specific modifications are as follows. The text of the revised draft has also been marked in red font:

The experimental materials used in this study were summer vegetable soybean pods produced in Tai 'an, Shandong Province, China. The pods were randomly laid on a white background board during capture, and then H680 scanner produced by Hanwang Tech-nology and HonorV30pro produced by Honor Technology of China were used to capture a total of 1072 images of the pods as original images.

5.Checking all manuscript for proper standardization.
Response: Thank you very much for your advice. We've checked to make sure the manuscript is up to standard.

6.Scientific names in italics;
Response: Thank you very much for your advice. We've already used scientific names in italics. Specific modifications are as follows:
Vegetable soybean [Glycine Max (L.) Merrill]

7.Some figures need improve quality (Figure 9)
Response: Thank you very much for your suggestion. We have redone Figure 9. At present, these results have changed due to the adjustment of the non-maximum suppression value. As for the seed count, the linear fitting graph of seed count was mainly modified, and the seed number obtained by the common classification method and the seed number obtained by our improved classification method were compared with the true value. See Figure 9 for details:

Figure 9. Analysis of seed counting results. (a) Correlation analysis between count results of common classification method and true value. (b) Correlation analysis between count results of improved classification method and true value. (Please check the figure within the attachment.)

8. L118-131. Its major rewrite;

Response: Thank you very much for your suggestion, we have rewritten this section on page 2 line 50 to page 3 line 5. The details are as follows, marked in red in the text of the revised manuscript:
The aim of this paper was to propose a fast and accurate pod detection method, which can solve the problem of counting pods and seeds in soybean indoor planting test. This method can not only realize the classification and counting of pods, but also distin-guish the effective grains and abortive grains inside the pod and count the effective grains separately. In this study, we first established a training dataset with 3216 images, and also made a distorted image test dataset, a high-density pods image test dataset, and a low-pixel image test dataset containing 90 images each. Then, four object detection models such as Faster R-CNN, YOLOv3, YOLOv4 and YOLOX were trained on the training da-taset, and the performance were compared on the three test datasets to select the best model. Finally, our pod classification and counting method was validated.

9. L142. 30 cm;
Response: Thank you very much for your suggestion and this error has been corrected.

10. y-axis and x-axis were need define.
Response: Thank you very much for your suggestion. The x-axis and y-axis have been defined.

11. Check old references for accuracy.
Response: Thank you very much for your suggestions. We have checked the old references to ensure their accuracy.

Reviewer 2 Report

The author has proposed “High-Throughput Classification and Counting of Vegetable Soybean Pods Based on Deep Learning”, it is an interesting topic, however, I have following comments.

1. Author should mention the dataset size (no of images) in the abstract.

2. Line 21: did you do any modification in the existing models, if yes then mention that.

3. The literature is not sufficient to cover the said area. Many latest references have not been mentioned in the literature. The author should include the latest literature in the manuscript and highlight their contribution. Some of the studies are as follows:

4. An Artificial Intelligence-Based Stacked Ensemble Approach for Prediction of Protein Subcellular Localiza6on in Confocal Microscopy Images; Enhancing Image AnnotationTechnique of Fruit Classification Using a Deep Learning Approach; Soybean seed counting based on pod image using two-column convolution neural network; Fruit Image Classification Model Based on MobileNetV2 with Deep Transfer Learning Technique.

5. Highlight (bullet points) the contribution of this article

6. Change the word “shoot” to capture in line 136, 139. (and other places)

7. Author did not mention the different classes of images, they mentioned they have taken 1072 images (137 line) and 710 images (line 139) but fail to show them in different classes.

8. It is must to show the no. of images in different classes, otherwise if it is just one class then there is no need for classification. Mention: in abstract as well as in section 2.1.1

9. There is the problem of limited data/data imbalance which I guess author hide it from the manuscript. How did author overcome this problem?

10. Show the no. of images per class in a tabular form, so it dataset details will be apparent

11. Section 2.2: how did author decide to choose those specific methods? Are there any criteria or something?

12. I guess there should be a model selection process, in which all state-of-art models should be run on the dataset, to find out which are the model that can perform better on the mentioned dataset.

13. Why they didn’t choose mobilenetv2, inception, densnet, alxnet and other models?

14. Authors have used R-CNN, YOLOv3, YOLOv4 and YOLOX for experiments but fail to mention weather they have used those models directly or there is any modification in the architecture.

15. Future directions are missing in the conclusion, and also it is must the author should mention the practical usage of this work in introduction as well as in conclusion.

16. How about sharing the model details (code) and dataset?

Author Response

Responses to the comments of Reviewer #2

1. Author should mention the dataset size (no of images) in the abstract.
Response: Thank you very much for your advice. We have already mentioned the data set size in the abstract (lines 20-21 on page 1 of the revised version). The specific modification is shown in red letters:

At the same time, this paper designs a pod dataset based on multi-device capture, in which the training dataset after data augmentation has a total of 3216 images, and the distortion image test dataset, the high-density pods image test dataset and the low-pixel image test dataset include 90 images respectively.

2. Line 21: did you do any modification in the existing models, if yes then mention that.
Response: Thank you very much for your advice. The model in this paper has not been modified. Our original intention is to focus on the application of pod recognition rather than the algorithm itself.

3. The literature is not sufficient to cover the said area. Many latest references have not been mentioned in the literature. The author should include the latest literature in the manuscript and highlight their contribution. Some of the studies are as follows:
4. An Artificial Intelligence-Based Stacked Ensemble Approach for Prediction of Protein Subcellular Localization in Confocal Microscopy lmages; Enhancing lmage Annotation Technique of Fruit Classification Using a Deep Learning Approach; Soybean seed counting based on pod image using two-column convolution neural network; Fruit lmage Classification Model Based on MobileNetV2 with Deep Transfer Learning Technique.

Response [3,4]: Thank you very much for your suggestions. The following three references have been added:
[7] Soybean seed counting based on pod image using two-column convolution neural network;
[15] An Artificial Intelligence-Based Stacked Ensemble Approach for Prediction of Protein Subcellular Localization in Confocal Microscopy lmages;
[17] Fruit lmage Classification Model Based on MobileNetV2 with Deep Transfer Learning Technique.

5. Highlight (bullet points) the contribution of this article
Response: Thank you very much for your advice. We have added an explanation of the contribution of this paper to lines 4-15 of the revised manuscript. The specific content added is as follows, and in red in the revised manuscript:
The main contribution of this study are as follows:
(1) We propose a new method for fine classification of pods based on the number of effective seeds and abortive seeds in pods. This method can obtain the number of effective seeds more accurately. To the best of our knowledge, this is the first work that can distinguish the effective seeds from the abortive seeds in soybean pods.
(2) A soybean pod dataset based on multiple devices is designed to simulate the problems of high pod density, image distortion and low resolution encountered in image acquisition in reality.
(3) The proposed object detection network achieves good performance when processing densely sampled soybean pods, and can effectively identify distorted images and low-pixel images.

6. Change word "shoot" to capture in line 136,139. (and other places)
Response: Thank you very much for your advice. The word "shoot" in the manuscript has been corrected to "capture".

7. Author did not mention the different classes of images, they mentioned they have taken1072 images (137 line) and 710 images (line 139)but fail to show them in different classes.
8. lt is must to show the no. of images in different classes, otherwise if it is just one class then there is no need for classification. Mention: in abstract as well as in section 2.1.1

Response [7,8]: Thank you very much for your advice. In order to better explain how we classify pod, we chose 2.1.2 data annotation in the previous manuscript to elaborate our classification method, so the classification diagram is placed in 2.1.2, while the acquisition of images was emphasized in 2.1.1 and the abstract. The specific classification diagram is shown in Figure 4 (please check the attachment).

9. There is the problem of limited data/data imbalance which l guess author hide
it from the manuscript.How did author overcome this problem.
Response: Thank you very much for your question. For data-constrained problems, our approach is to try to balance the class distribution as much as possible by randomly repeating samples from minority classes.

10. Show the no. of images per class in a tabular form, so it dataset details will be apparent
Response: Thank you very much for your advice. We added Table 2 in section 2.1.2 to show the number of pods in the dataset
Table 2. Distribution of each class of pods in different datasets. (Please check this table within the attachment).

11.Section 2.2: how did author decide to choose those specific methods? Are there any criteria or something?
Response: Thank you very much for your question. In the previous experiment, we divided pods into one pod, two pods, and three pods for classification and counting, and some models did not perform well, so we chose the model with better effect in this experiment. And we have added the reasons for choosing these models in the revised manuscript on page 6, lines 17-19, Section 2.2.1. The specific additions are as follows, marked in red letters in the revised manuscript:
In previous work, we have divided pods into one-seed pod, two-seed pod, and three-seed pod, and then compared the recognition effects of SSD, YOLOv3, YOLOv4, and YOLOX, where which SSD has a poor effect. Considering the recognition accuracy and recognition speed, four object detection algorithms Faster R-CNN, YOLOv3, YOLOv4 and YOLOX were selected for comparative experiments under the same experimental environment. The four algorithms were compared in many aspects through performance evaluation indicators, and finally the algorithm with the best comprehensive performance was selected as the algorithm for pod classification and counting.

12. l guess there should be a model selection process, in which all state-of-art models should be run on the dataset, to find out which are the model that can perform better on the mentioned dataset.
Response: Thank you very much for your advice. A flowchart has been re-added to the revised manuscript Section 2.2.2 to highlight the process of model selection, and the process of model selection is further elaborated on page 6, lines 26-31. The details are as follows:
Figure 5 illustrates the construction process of the pod classification and counting model. Then, three test datasets are used to test the performance of the constructed model. By comparing the performance indicators of different models, the model with the best comprehensive performance is finally selected as the model of pod classification and counting. Through this model, the effective pod number, the effective seed number and the number of
each kind of pod can be obtained quickly and accurately.

Figure 5. The key flow of pod classification and counting model construction. (Please check this figure within the attachment).

13. Why they didn't choose mobilenetv2, inception, densnet, alxnet and other
models?
Response: Thank you very much for your question. The reason for not choosing other models is that as shown in the answer to the second suggestion, the object detection model we use is not modified, so the object detection network we use is still the original backbone network, such as the backbone network of YOLOX is CSPdarknet53.

14.Authors have used R-CNN,YOLOv3,YOLOv4 and YOLOX for experiments
but fail to mention weather they have used those models directly or there is any
modification in the architecture.
Response: Thank you very much for your advice. As stated in the answer to the second suggestion, the model used in this paper has not been modified. We give an explanation on line 20-21 on page 6 of the revised manuscript. The specific content added is shown below and marked in red font in the revised manuscript:
Considering the recognition accuracy and recognition speed, the original network models of four object detection algorithms Faster R-CNN, YOLOv3, YOLOv4 and YOLOX were selected without modification for comparative experiments under the same experimental environment.

15.Future directions are missing in the conclusion, and also it is must the author should mention the practical usage of this work in introduction as well as in conclusion.
Response: Thank you very much for your advice. We have explained the practical usefulness of this paper in lines 50-51 on the first page of the revised manuscript as well as lines 4-8 on page 17, while the future directions are presented in lines 13-15 on page 17. The details are as follows:

The aim of this paper was to propose a fast and accurate pod detection method, which can solve the problem of counting pods and seeds in soybean indoor planting test.
Experimental data show that our high-throughput pod classification and counting method speeds up laboratory testing of soybeans, solves the problem of pod and seed counting in laboratory testing of soybeans, and reduces the errors of pod and seed count-ing, which is helpful to improve current and future breeding programs.

In the future research, we intend to further explore a more accurate and convenient method to directly detect the complete soybean plant outdoors to obtain the category and number of pods.

16.How about sharing the model details (code) and dataset?
Response: Thank you very much for the recognition of our research work by your suggestion. However, due to our laboratory policy or confidentiality agreement, we cannot provide the original data. We have fully described the experimental design, analysis, and results, as well as the process of data analysis and processing. If you have questions about specific data, we will do our best to provide more detailed explanations and clarifications.

Reviewer 3 Report

The paper presents an approach to counting vegetable soybean pods based on deep learning. The article needs a revision of English language and style, and other adjustments to improve its quality.

Major revision:

1)      A discussion of the usefulness of the proposed method for a real application is lacking, as pods are counted using a white background. Is this condition feasible to be realized in a real application?

2)      At the end of the Introduction, add a paragraph about paper structure.

3)      Page 4: What were the values of the parameters used in the data augmentation? The authors could have used other data augmentation methods, such as mirror image, rotation, noise, scale, etc.

4)      Section 2.1.2: Inform the distribution of classes in the datasets.

5)      Section 2.2.4: There should be a metric to measure the ratio between the number of identified seeds and the actual number.

6)      Page 9: “that the models are less affected by color distortion” --> The data augmentation method (page 3, lines 142 to 148) may have helped with this result. Perhaps the use of other data augmentation methods, as mentioned earlier, can help with other shortcomings of the models. Comment this in the text.

7)      Page 9: “the test dataset HPD are taken by different cameras” --> On page 5 is mentioned: “pods images were taken by Huawei P40 as the test dataset HPD”. Was one camera used or several different cameras?

8)      Page 11: Comment on Table 5.

9)      Figure 8: Apparently, YOLOX detects more pods than the actual number. You should check if it is possible to adjust the non-maximal suppression parameters. I believe it can improve the results.

10)   Page 14: “The traditional classification and counting method” --> What is the traditional method?

11)   Page 14: “the average error was 4.39, and the average relative error was 5.57%”  --> The result on page 13 is different.

12)   Datasets and codes could be made available.

Minor revision:

Abstract: marked to make a dataset. --> marked to make datasets.

Abstract: detection models [Faster R-22 CNN, YOLOv3, YOLOv4, YOLOX] is --> detection models, Faster R-22 CNN, YOLOv3, YOLOv4, and YOLOX, is

Abstract: and 90.27% respectively in --> and 90.27%, respectively, in

Page 1: [Glycine Max (L.) Merrill] --> It is a reference? Fix it.

Page 2: Add the meaning of SPPOD and QTLS.

Page 2: extraction (FE), then the support --> extraction (FE), the support

Page 2: machine vision method. --> machine vision methods.

Page 3: one pod, two pod, three pod, four pod and five 99 pod, and used --> one pod, two pods, three pods, four pods, and five pods, and used

Page 3: branch Angle --> branch angle

Page 3: and 91% respectively. --> and 91%, respectively.

Page 4: which is the number of data sets --> which is the number of samples in the datasets

Page 6: Figure 6 shows --> Figure 5 shows

Page 8: Precision, recall --> Precision, Recall

Page 8: model. precision is --> model. Precision is

Section 3: Write the text in this section in the past tense because they have already been done. Some examples: experiment adopts --> experiment adopted; iterates 150 --> iterated 150; and freezes the --> and froze the; etc.

Page 9: the migration learning --> the transfer learning

Page 9: Table 2, Table 3, and Table 4 show --> Tables 2, 3, and 4 show

Page 11: and three pods --> and three-seed pods

Figures 8 and 9 are actually Figures 7 and 8, respectively.

Page 13: problems even if the pod --> problems even with the pod

Page 14: in the table 9. --> in Table 9.

Page 16: The purpose of this study is to --> The purpose of this work was to

Author Response

Please check the replies within the attachment.

Round 2

Reviewer 1 Report

I recomend accept in present form these manuscript. 

Reviewer 2 Report

Author has addressed all the comments 

Reviewer 3 Report

Congratulations! The article obtained merit to be published. I am happy to have contributed a little to the article.